# Next Generation Sequencing of Cerebrospinal Fluid B Cell Repertoires in Multiple Sclerosis and Other Neuro-Inflammatory Diseases—A Comprehensive Review

**DOI:** 10.3390/diagnostics11101871

**Published:** 2021-10-11

**Authors:** Christoph Ruschil, Constanze Louisa Kemmerer, Lena Beller, Gisela Gabernet, Markus Christian Kowarik

**Affiliations:** 1Department of Neurology & Stroke, Eberhard-Karls University, 72076 Tübingen, Germany; christoph.ruschil@uni-tuebingen.de; 2Hertie Institute for Clinical Brain Research, Eberhard-Karls University, 72076 Tübingen, Germany; constanze.kemmerer@uni-tuebingen.de (C.L.K.); lena.beller@student.uni-tuebingen.de (L.B.); 3Quantitative Biology Center (QBiC), Eberhard-Karls University of Tübingen, 72076 Tübingen, Germany; gisela.gabernet@qbic.uni-tuebingen.de; 4Department of Neurology, Klinikum Rechts der Isar, Technische Universität München, 81541 Munich, Germany

**Keywords:** B cells, B cell repertoire, CSF, NGS, next-generation sequencing, multiple sclerosis, MS, NMOSD, limbic encephalitis

## Abstract

During the last few decades, the role of B cells has been well established and redefined in neuro-inflammatory diseases, including multiple sclerosis and autoantibody-associated diseases. In particular, B cell maturation and trafficking across the blood–brain barrier (BBB) has recently been deciphered with the development of next-generation sequencing (NGS) approaches, which allow the assessment of representative cerebrospinal fluid (CSF) and peripheral blood B cell repertoires. In this review, we perform literature research focusing on NGS studies that allow further insights into B cell pathophysiology during neuro-inflammation. Besides the analysis of CSF B cells, the paralleled assessment of peripheral blood B cell repertoire provides deep insights into not only the CSF compartment, but also in B cell trafficking patterns across the BBB. In multiple sclerosis, CSF-specific B cell maturation, in combination with a bidirectional exchange of B cells across the BBB, is consistently detectable. These data suggest that B cells most likely encounter antigen(s) within the CSF and migrate across the BBB, with further maturation also taking place in the periphery. Autoantibody-mediated diseases, such as neuromyelitis optica spectrum disorder and LGI1 / NMDAR encephalitis, also show features of a CSF-specific B cell maturation and clonal connectivity with peripheral blood. In conclusion, these data suggest an intense exchange of B cells across the BBB, possibly feeding autoimmune circuits. Further developments in sequencing technologies will help to dissect the exact pathophysiologic mechanisms of B cells during neuro-inflammation.

## 1. Introduction

### 1.1. Neuro-Inflammatory Diseases—Cerebrospinal Fluid (CSF) Findings, Including Routine Diagnostics, Autoantibodies and B Cells

B cells are an essential part of the adaptive immune system and play important roles in the pathogenesis of several neuro-inflammatory diseases. The functional properties of B cells are manifold and not only include antigen recognition through B cell receptors (BCR) and specific antibody production, but also antigen presentation to other immune cells and the secretion of cytokines [1]. Central nervous system (CNS) inflammation is often modulated or even initiated by B cells. The priming of these B cells can either occur in the peripheral compartment, followed by trafficking into the CNS, or by a compartmentalized B cell reaction within the CNS [1]. There are several pathways by which B cells could enter the CNS, including the blood–brain barrier (BBB), blood–cerebrospinal fluid (CSF) barrier or barriers at the surface of the human brain [2]. Critical questions regarding B cell pathophysiology within the CNS include immune-tolerance, the location of antigenic stimulus and antigenic stimulation itself, the composition of migratory B cell populations, intrathecal B cell maturation and the re-circulation of B cells into the periphery.

B cells are consistently found in the CSF during neuro-inflammation in both infectious and autoimmune diseases but are largely absent under non-inflammatory conditions [3,4,5]. Among autoimmune CNS disorders, MS is the most common disease and patients present with various neurological deficits. The initial disease phase is often characterized by relapses, and in a substantial proportion of MS patients, is followed by a progressive disease phase [1,6]. It is believed that an orchestrated T and B cell reaction leads to inflammation in the CNS, resulting in demyelination and neuronal damage; this disease phase is often followed by compartmentalized CNS inflammation and neuro-degeneration [1]. The presence of oligoclonal bands (OCBs) as a consequence of an intrathecal B cell activation has been described since the 1940s and still serves as a diagnostic hallmark [7,8]. Furthermore, B cell percentages are elevated in the CSF of MS patients; CSF plasmablasts represent the most frequent antibody-secreting B cell subset correlating with intrathecal IgG production and disease activity [9]. Nevertheless, a defined antigen target for CSF antibodies has not been consistently identified in MS yet. In contrast to MS, NMOSD constitutes a B cell-mediated neuro-inflammatory disease that has been characterized by pathogenic antibodies targeting the water channel aquaporin 4 (AQP4-IgG) [10,11,12]. NMOSD mainly affects the spinal cord and optic nerves, which can lead to similar clinical presentations, as in MS patients [13,14]. AQP4-IgG has been shown to target astrocytes, which show a high expression of AQP4 channels, subsequently leading to astrocyte injury and demyelination [14]. Disease-relevant AQP4-IgG antibodies are detectable in serum [10,15] as well as CSF [16]; however, the level of intrathecal IgG synthesis is low, transient and mostly restricted to acute relapses [17]. CSF oligoclonal IgG bands have been infrequently observed in NMOSD [17]. Regarding CSF white cell counts, more than 50% of CSF samples show an elevated cell count with mostly lymphocytes and monocytes [18]. The effectiveness of B cell-depleting therapies in NMOSD and MS [19,20,21,22,23,24,25,26] has underlined the importance of B cell-mediated autoimmunity in both diseases. Concerning other autoantibody-mediated encephalopathies, numerous antibodies targeting neuronal or glial cells have recently been described in neurological diseases. In this context, the anti-N-methyl-d-aspartate receptor (NMDAR) and leucine-rich, glioma-inactivated 1 (LGI1) autoantibody-mediated encephalitis is one of the best-characterized disease entities [27]. Although CSF LGI1 antibodies are detected in around 90% of patients, there is an infrequent association with CSF lymphocytosis and OCB in LGI1 encephalitis [28,29]. Regarding NMDAR encephalitis, NMDAR autoantibodies are consistently detectable within the CSF compartment while around 14% of the patients are serum-negative [30]; an elevated CSF cell count is found in around 74% and positive OCBs in approximately 70% of the patients [31]. From a clinical perspective, patients with autoantibody-mediated encephalitis often present with new-onset psychosis, amnesia, hyperkinesia or vegetative dysfunction. With the discovery of anti-neuronal antibodies, these patients now receive causal therapies, including B cell depletion and antibody removal, instead of solely symptomatic treatments [27].

Accordingly, there are multiple lines of evidence indicating that B cells modulate or initiate inflammatory diseases in the CNS in several cases with defined antigen targets. Although inflammatory antibody-mediated cascades have been defined, the initial events that lead to B cell autoimmunity, the place of initial B cell priming and further B cell trafficking/maturation, must be fully determined. With the development of new high-throughput sequencing techniques, there have been huge advances in tracking B cell development and migration through the assessment of B cell repertoires in the CSF and peripheral blood (PB) compartment on the transcriptome level. The aim of this review is to summarize these developments and extract the major findings on B cell maturation and migration in neuro-inflammatory diseases. We first provide a short background on the biological context of B cell sequencing and the approach of repertoire sequencing, and then we explain the methods of this review. Next, we summarize major studies on B cell receptor sequencing in a chronological order and discuss the major findings of the studies mentioned.

### 1.2. B Cell Repertoires—The CDR3 Region as a Molecular Fingerprint of B Cell Maturation

B cells develop from hematopoietic stem cells in the bone marrow, where various steps in the assembly and expression of functional B cell receptor (BCR) genes take place [1]. During this maturation process, a primarily huge diversity of B cells is generated by somatic recombination of the variable (V), diversity (D) and joining (J) gene segments (variable heavy (VH) and light (VL) chain immunoglobulin loci) that form the CDR3 antigen-binding region. At this immature B cell stage, cells are first tested for tolerance to self-antigens (central tolerance); if cells are non-reactive, they leave the bone marrow. In the periphery, B cells become activated through antigen binding of the B cell receptor in addition to stimulating signals. At this stage, BCR genes can be further diversified through somatic hypermutation (SHM) in germinal center reactions and/or by undergoing a class switch recombination [1]. Somatic hypermutations occur by introducing point mutations in the variable region, which lead to affinity maturation towards a defined target and changes in the CDR3 [32]. Contrastingly, these mutations can drive the formation of auto-reactive B cells into the periphery. However, mechanisms including deletion of auto-reactive B cell clones and suppression of B cells through anergy and immunomodulation with regulatory T and B cells can further prevent auto-reactivity (peripheral tolerance). Altogether, these maturation steps lead to an exceptional diversity of the peripheral B cell repertoire and offer the possibility to trace back B cell maturation according to SHM patterns. In the context of neuro-inflammation, B cell maturation and trafficking in and out of the inflamed CNS can be studied by analyzing the CDR3 region of BCR repertoires [33]. With the development of next-generation high-throughput sequencing methods, it is possible to assess up to millions of immunoglobulin/BCR transcripts, including the CDR3 region, and to actually sequence representative B cell repertoires, not only from the CSF but also from peripheral blood B cells. Therefore, Ig gene transcripts, including the CDR3 section of heavy chain variable regions (Ig-VH), provide molecular fingerprints that permit temporal and spatial tracking of clonally related B cells [34].

### 1.3. Next-Generation Sequencing of B Cell Repertoires—Methods and Data Analysis

The development of next-generation sequencing technologies has made an impact on all fields of genomics, including BCR repertoire sequencing (Rep-seq). Departing from Sanger sequencing protocols, technologic progress has evolved to sequencing millions of BCR sequences in parallel [35]. B cell repertoires can be generated from PBMCs or CSF cells by whole RNA extraction. By adding a sorting step via flow cytometry prior to RNA extraction, different B cell populations can be distinguished within a repertoire. In current high-throughput methods, the specific amplification of the BCR genes can be achieved by PCR of the genomic DNA (gDNA) or immunoglobulin mRNA. Several protocols can be used for amplification, including multiplex PCR with primer pairs for the different VH and constant regions, and 5’ rapid amplification of cDNA ends (RACE) PCR [36]. The incorporation of unique molecule identifiers (UMI) as part of the sequencing primers allows for PCR sequencing error and amplification bias correction [37]. Barennes et al. recently showed that UMI-based methods are more accurate in quantifying the abundance of specific clonotypes, whereas non-UMI 5’RACE methods show a higher sensitivity [38,39].

With the possibility of sequencing up to several million Ig transcripts in parallel, a need for bioinformatics pipelines evolved in order to handle the high-throughput data analysis. The approach for an immunoglobulin repertoire data analysis generally involves filtering the sequencing reads according to quality criteria, building a consensus sequence to minimize errors introduced during library preparation and amplification, assigning V(D)J gene segment alleles by aligning to germline BCR databases and clonotyping and characterizing the repertoire [40]. Multiple bioinformatics tools and comprehensive pipelines have been developed to process B cell repertoire data [41,42].

Two popular tools used for aligning the reads to germline references are IgBlast [43]—based on the Blast algorithm—and IMGT/HighV-Quest [44], based on the Smith–Waterman algorithm. These are typically used to align to the ImMunoGeneTics (IMGT) germline database [45] and are included in several frameworks and pipelines [46] as well as individual alignment algorithms [47].

Clonotyping is another critical step in the repertoire analysis process, and the choice of tool has a big impact on the downstream repertoire characterization. Whereas some tools define a clonotype as identical sequences [48] that will generate a unique antibody, others allow clonotype grouping of similar (but not identical) sequences [46], typically by calculating nucleotide or amino acid sequence similarities or employing specific substitution models [46]. Once the sequence data is processed and the clonotypes defined, a quantitative characterization of the repertoires (e.g., repertoire diversity [49]) is needed to allow comparisons among individuals and experimental conditions.

## 2. Methods of Systematic Review

We performed a systematic database search in accordance with the preferred reporting items for systematic reviews and meta-analyses (PRISMA) guidelines. All published reports and articles were searched and accessed in May 2021 on the pubmed database. The following search terms were used: ((“cerebrospinal fluid” NOT “CSF-GM”) AND(“B cell repertoire” OR (“BCR” NOT “BCR-ABL”)) OR ((“cerebrospinal fluid” NOT “CSF-GM”) AND (“NGS” OR “next-generation sequencing”) AND “B cell”) OR((“cerebrospinal fluid” NOT “GM-CSF”) AND “B cell” AND “sequencing”). Papers were first screened on titles and abstracts (*n* = 135) to exclude papers with non-neurological purposes such as lymphoma, glaucoma or leukemia. Preselected articles (*n* = 75) were read in full-text. Articles were included in the systematic review using the following criteria: (1) the study included data on human research and not animal research, (2) the study displayed original data and did not report previously published data, or was a review of previous work or case report (reported patients *n* = 1) and (3) the study methods stated sequencing of the B cell receptor. This search and inclusion criteria are described in the flowchart (Figure 1) and resulted in 38 articles selected for further analysis for the comprehensive review. In addition, we examined papers cited in the selected articles and included additional references based on their relevance regarding the scope of this paper. Due to the very inhomogeneous methodological approach between various studies (e.g., sequencing methods/platforms, CSF B cell subtypes studied and bioinformatics data processing used, patient selection) and limited sample sizes in the studies mentioned, no statistical meta-analysis could be conducted.

## 3. Results

In order to learn more about B cell maturation and trafficking in neuro-inflammatory diseases, we performed the aforementioned search strategy and found several studies on B cell repertoire mass sequencing in multiple sclerosis and single studies on NMOSD as well as LGI1 and NMDAR autoantibody-mediated encephalitis. Early studies in MS with conventional B cell repertoire sequencing in the CSF compartment were also included in the analysis, in order to compare between the different methods and complete the picture of B cell pathophysiology in MS. B cell repertoire mass sequencing results in complex datasets; for this reason we ordered the results according to findings from basic repertoire analysis, clonal expansion within the CSF, B cell trafficking across the BBB and—if available—longitudinal CSF studies on single MS patients.

### 3.1. CSF B Cell Repertoires in Multiple Sclerosis

The presence of oligoclonal bands (OCBs) still serve as important criteria in establishing the diagnosis of multiple sclerosis [8]. Furthermore, OCBs and an intrathecal Ig synthesis have consistently been associated with elevated CSF B cell counts [9]. During the last few decades, numerous studies further examined the role of CSF B cells by assessing B cell repertoires within the CSF, in lesions of MS patients [50] and—more recently—in draining cervical lymph nodes [51]. Although studies have not provided defined targets of CSF antibodies in MS yet, a deeper understanding of B cell trafficking, B cell maturation and compartmentalized B cell reactions could be accomplished.

#### 3.1.1. Enrichment of VH4 Family Usage in CSF B Cell Repertoires 

The basic analysis of B cell repertoire properties in the CSF of MS patients indicates that the VH germline usage of CSF B cells differs from the expected germline prevalence. Most studies on CSF B cell repertoires show a consistent shift towards an increased VH4 family usage (Table 1). In particular, the VH4 family members VH4-31, VH4-34, VH4-39, IGHV4-59 and IGHV4-61 [52,53,54,55,56], have been utilized in CSF B cell clones. In this context, CD138+ plasmablast/-cells show a more pronounced bias towards VH4 family usage than CD19+ B cells [52]. An over-representation of VH4 in combination with VH2 families has been discussed in clinically isolated syndromes [57]. These findings are in line with results from B cell repertoires derived from CNS MS lesions, which also found an over-representation of VH4 families [50,58,59,60,61], including VH4-34 and VH4-39 [51,58] family members. In addition, an over-representation of some VH1 family member shifts in VHD and VHJ segments has also been reported occasionally in MS lesions [58]. It has been suggested, that the CSF VH4 family bias might be applied to distinguish multiple sclerosis from other neuro-inflammatory diseases [62] as well as predict the development of MS in patients with clinically isolated syndrome (CIS) [57].

#### 3.1.2. Clonal Expansion of B Cells within the CSF Compartment

As outlined above, B cell receptors (BCR) show a high variability due to the recombination of V-(D)-J segments, affinity maturation and class-switch recombination as well as somatic hypermutations (SHM). As a consequence, B cell immunoglobulin (Ig) transcripts that contain the same V-(D)-J segments as well as highly similar SHM-profiles, notably in the CDR3 region, are considered to be clonally related. Utilizing the CDR3 region as a clonal marker, the patterns of SHM in related B cells can be further examined and their maturation trees assessed in order to follow B cell maturation steps. Furthermore, the abundance of different clones, diversity and degree of the SHM of the acquired B cell repertoires provide further evidence for compartmentalized maturation processes [46,63,64,65,66,67].

Clonal expansion and somatic hypermutations of CSF B cell repertoires are consistent features that have already been reported in early MS studies [52,53,57,68,69,70,71,72] and did not include a sufficient analysis of peripheral blood (PB) B cell repertoires. Additional analyses on the clonal overlap between B cell receptor sequences in MS brain parenchyma, meningeal lymphoid follicles and CSF further indicate an immunological continuum inside the CNS/CSF [72,73,74]. With the availability of high-throughput sequencing, it is now possible to assess representative B cell repertoires from peripheral blood and to test whether B cell clones are CSF specific (Table 1). Several studies have shown that CSF lineage trees exist exclusively in the CSF without clonal connection to the peripheral blood [33,54,56,62,75,76,77], indicating that B cell maturation and expansion is indeed occurring in the CSF compartment. This finding is indirectly supported by the disproportionate increase of all B cells during active MS and the overall increased number of antigen-experienced B cells in the CSF, assessed by flow cytometry [9,76]. When looking at B cell subsets within CSF clones in general, it could be shown that memory B cells, Ig27-IgD-double negative (DN) B cells and especially plasmablasts, contribute to CSF clonal groups [33]. Another study found alterations in the SHM profiles of CSF B cell subsets [56]. While the SHM profiles of CSF IgG transcripts appear to be similar to IgG transcripts in PB from switched memory B cells and plasma cells, SHM profiles of CSF IgM transcripts seem to be most similar to PB IgM transcripts of naive, unswitched memory and DN B cell subsets. Regarding Ig isotype switching in CSF clones as a feature of maturation processes, one study could not find evidence for an intrathecal isotype switching from the IgM to IgG subclass [55]. In contrast, another study found CSF Ig-VH clusters that were exclusively IgG or IgM as well as mixed clusters [34].

When comparing MS B cell repertoires in the CSF with control groups, early studies indicate that a nonrandom distribution of B cell clones in the CSF is not a distinctive feature of MS, but also occurs in a variety of infectious or autoimmune disorders [52,55,62,68,69,78]. Furthermore, it has been proposed that CSF B cell repertoires from MS samples contain more clonally-related sequences and significantly more SHM when compared with other neurological controls, including inflammatory diseases [54,62]. In sum, early studies as well as the more recent NGS studies on CSF and PB B cell repertoires both confirm that affinity maturation of B cells occur in the CSF compartment in multiple sclerosis.

As mentioned above, clonally-expanded B cells can not only be detected in the CSF and parenchymal infiltrates, but also in meningeal lymph-like follicles/aggregates of mainly progressive MS patients [73]. Interestingly, the majority of expanded antigen-experienced B cells derived from the meninges, were also present in the parenchyma. Altogether, clonal relationships can be established between different compartments including the meninges, inflammatory MS plaques and the CSF compartment.

#### 3.1.3. Spatial Tracking of B Cell Clones over the BBB—Bidirectional Exchange of B Cells 

By applying NGS sequencing, it has become possible to representatively sequence both CSF and peripheral blood (PB) B cell repertoires and thus examine B cell trafficking across the BBB. Overall, a substantial clonal overlap between peripheral blood and CSF B cells can be established in different studies (Table 1). While an early study found a restricted pool of clonally-related CSF B cells (on average 6.3%) that connect with peripheral blood, around one-third of CSF B cells showed a clonal connection with peripheral B cells in a later study [33,54]. After further observing lineage trees and the occurrence of SHM, B cell trafficking patterns across the BBB and the compartment of major clonal expansion can be further approximated. Although lineage trees between CSF and peripheral blood B cells seem to primarily undergo active diversification in the CSF compartment, B cells clones also acquire additional mutations in PB [33,54,55,56,62]. Bicompartmental lineage trees that demonstrate further maturation in the PB compartment can either represent parallel maturation steps of a common ancestor B cell in both compartments or indicate that CSF B cells recirculate into PB, where further maturation steps occur. A possible efflux of B cells out of the CSF is further supported by an interesting study that established B cell lineage trees between CNS lesions and B cells in draining cervical lymph nodes from MS patients [51]. Moreover, this study provided evidence that B cells migrate freely across the tissue barrier with substantial B cell maturation occurring outside of the CNS [51]. When looking at the B cell subtypes in PB that contribute to lineage trees between PB and CSF, switched memory B cells, double negative B cells and plasmaplasts/cells were first identified in order to connect with CSF B cells [56]. Further studies also observed a limited overlap between PB naïve B cells/unswitched memory B cells and CSF B cells [33,76]. Regarding treatment effects on B cell trafficking across the BBB, natalizumab diminished the exchange of PB and CSF B cells according to its well-established mode of action, whereas fingolimod did not seem to substantially interact with the B cell exchange [33].

In summary, these data indicate that primarily antigen-experienced PB cells enter the CNS/CSF with further compartment-specific maturation taking place within the CSF. Furthermore, it has been suggested that CSF B cells also recirculate in the peripheral blood, possibly through draining cervical lymph nodes, where presumed further maturation and interaction with other immune cells takes place.

#### 3.1.4. Temporal Tracking of B Cell Clones within the CSF Compartment: Treatment Specific Effects on B Cell Clones

Despite the exchange of B cells across the BBB, B cell repertoire sequencing and the generation of lineage trees also allows us to study clonal groups over time. Limited data on the clonal persistence of CSF B cells is available regarding untreated MS patients (Table 2); only one out of three patients showed one persistent B cell clone over time [34]. Regarding treated MS patients, clonal persistence was evaluated in 16 patients receiving natalizumab (*n* = 7), fingolimod (*n* = 5), interferon-beta (*n* = 3) or dimethyl fumarate (*n* = 1) within three different studies (Table 2). It is worth mentioning that we did not find any persistent clones in fingolimod-treated patients in our study but did find some huge persistent clones under the natalizumab treatment [33]. Another study reported persisting clonal populations in one out of two individuals for the latter two treatments [34]. However, these two patients with persistent clonal groups showed clinical and MRI activity over an observation period of more than 12 months, which contrasts with our patients, who were all clinically and radiologically stable over 6 months [33,34]. Regarding other treatments, no final conclusions can be drawn from the different studies since patients showed an inhomogeneous disease activity and studies had very low patient numbers (Table 2). In summary, these data suggest that CSF-compartmentalized immune reactions can be partially therapy resistant and also underline the need for further studies in homogeneous patient collectives.

#### 3.1.5. Overlap between Ig Transcriptome and Proteome: Evidence for CSF B Cells as the Origin of Intrathecal Ig

Intrathecal IgG synthesis is associated with disease severity and progression in multiple sclerosis [79,80]. Therefore, it appears relevant to determine whether and to what extent CSF B cells contribute to the formation of CSF OCBs. It has been observed that the Ig transcriptome of CSF B cells or brain lesions and Ig proteome of intrathecal OCBs show a relevant overlap [7,74,81]. Furthermore, peptides derived from intrathecal Ig can be matched to expanded CSF B cell clusters [62,75,77] and the matching IgG fraction mostly persisted in equal amounts in a longitudinal analysis [77]. Furthermore, dominant CSF B cell clones that match with CSF Ig peptides can consistently be identified in the PB compartment [62,75,77], indicating that OCBs are not only the result of a targeted immune response within the CSF but represent an active B cell immune reaction that is presumably supported on both sides of the blood–brain barrier [75]. This finding is supported by another study [82] showing that antibodies derived from OCB-related CSF B cells have conformational epitopes of ubiquitous intracellular proteins that are not specific to brain tissue.

### 3.2. CSF B Cell Repertoires in Autoantibody-Mediated Encephalitis

During the last two decades, several autoantibodies have been identified to initiate and maintain autoimmune encephalitis. Regarding the pathophysiological mechanisms of AQP4-IgG-mediated inflammation, clinical and experimental data indicate that AQP4-IgG activates the classical complement cascade in NMOSD, which is believed to initiate astrocyte injury [83]. Different pathogenic mechanisms were proposed for NMDAR and LGI1 autoantibodies; where NMDAR autoantibodies lead to receptor cross-linking, internalization and degradation, resulting in a reduced number of NMDARs on the neuronal surface, LGI1 autoantibodies induce neuronal dysfunction by interrupting the trans-synaptic binding of LGI1 to its receptor ADAM22 [27]. Whether disease-relevant antibodies are produced in the peripheral blood, leaking through the BBB or within the CNS compartment by intrathecal B cell clones, remains a key question in the pathophysiologic understanding of autoantibody-mediated encephalitis.

#### 3.2.1. Neuromyelitis Optica Spectrum Disorders

The first evidence for an intrathecal production of AQP4-IgG by CSF B cells was found in a study applying single-cell sorting and RT-PCR in one NMOSD patient [16]. B cell repertoire analysis in this study and a successive study including seven NMOSD patients [5] revealed a dynamic, clonally-expanded plasma cell population with features of an antigen-targeted response. The majority (around 65%) of produced recombinant antibodies from clonally-related CSF B cells were AQP4 specific [5,16]. A paralleled analysis of peripheral blood B cell populations applying next-generation sequencing revealed that CSF B cells seem to connect with memory B cells, DN B cells and plasmablasts, with DN B cells showing an intensive clonal relation to AQP4-specific CSF B cells [5]. By overlapping the recovered Ig transcriptome libraries from CSF and PB with Ig proteomics in both compartments, it could be observed that a proportion of CSF AQP4-IgG is indeed produced locally while serum AQP4-IgG might leak additionally through an open BBB [5,17]. Another study in NMOSD showed that sequences derived from a common clone could be detected in plasmablasts on both sides of the BBB [84]. In summary, multiple lines of evidence suggest that the primary autoimmune response against AQP4 initiates with the release of AQP4-specific memory B cells, DN B cells and plasmablasts in peripheral blood, while some B cells undergo clonal expansion and affinity maturation in the CSF compartment [5].

#### 3.2.2. LGI1 and NMDAR Autoantibody-Mediated Encephalitis

By applying next-generation sequencing of CSF and PB B cell repertoires, one study found indirect evidence for a CNS-based antigen-driven response in six patients with LGI1 antibody encephalitis [28]. CSF B cell repertoires comprised a discrete number of highly expanded clusters with intensive mutational activity, which points towards an intrathecal maturation process. Data from another study suggest that 84% of the ASCs and 21% of the memory B cells in LGI1 CSF encode for LGI1-reactive antibodies and display a high degree of somatic hypermutations [85]. Regarding the connectivity of CSF clones to the peripheral blood compartment, a marked clonal overlap between CSF B cells and switched memory, plasmablasts and plasma cells was detectable [28].

Regarding NMDAR antibody encephalitis, one study applied a B cell single-cell approach to examine the CSF B cell repertoire with regard to a specific, targeted antibody production [86]. Surprisingly, only around 6% of antibody-secreting cells produced NMDR specific antibodies, although all patients showed an intrathecal synthesis of NMDAR antibodies. However, most of the other CSF B cell-derived antibodies reacted against other brain-expressed epitopes, including neuronal surface antigens in the hippocampus and cerebellum. Both subsets of antibody-producing cells showed features of an intrathecal clonal expansion but a relatively low number of somatic hypermutations [85,86]. Another study found a preferential usage of certain germline segments in CSF clones and several shared clones between different individuals applying single-cell sequencing and RT-PCR [87]. To the best of our knowledge, shared clones between different individuals have not been reported in other studies and diseases, so this finding might be restricted to NMDAR antibody encephalitis or result from methodological problems.

In summary, an intrathecal B cell maturation was evident in all autoantibody-mediated CNS diseases examined, suggesting that a compartmentalized B reaction and clonal connection with PB is a common feature of CNS-specific humoral immune responses.

## 4. Discussion

During the last few decades, the role of B cells has been well established and redefined in neuro-inflammatory diseases, changing the pathophysiological understanding in multiple sclerosis and other disease entities. The development of high-throughput next-generation sequencing has further contributed to push the boundaries of B cell immunology and helped to representatively assess peripheral blood B cell repertoires, in addition to the CSF compartment, thus allowing insights in B cell maturation processes and trafficking patterns across the blood-brain barrier.

Although several studies applying NGS have been performed on multiple sclerosis with mostly redundant results, studies suffer in several methodological issues: (1) the number of studied patients is low due to the high effort and costs, (2) MS patients show partially different disease states, which sometimes makes it difficult to compare between patients and studies, (3) early studies failed to use a unique molecular identifier, which helps to minimize over-amplification and sequencing errors, (4) bioinformatics sub-sampling has not been performed in most studies, in order to minimize effects resulting from a different inter-individual sequencing depth, and (5) mass sequencing only allows a snapshot of the B cell repertoire at a certain time and does not fully represent the dynamic relationship among transiting B cells. Nevertheless, the following conclusions can be drawn from the mentioned studies:Regarding the analysis of basic B cell repertoire characteristics, a preferential usage of the VH4 germline family within CSF B cell repertoires has been consistently shown in several studies. These data suggest that a chronic B cell stimulation with a common mechanism or antigen(s) might occur with epitopes that are preferentially recognized by the hypervariable loop structure of VH4 segments.These results go along with the consistent finding from both single cell and NGS CSF B cell repertoire analyses, indicating that a substantial proportion of CSF B cell clones undergo CSF compartment-specific maturation steps, including clonal expansion.The most striking results derived from NGS studies on CSF and PB B cell repertoires relate to B cell trafficking across the BBB. An intense exchange of B cells across the BBB is observed in MS and other autoimmune disorders, pointing towards an involvement of additional compartments outside of the CNS. Studies consistently show that B cells not only migrate into the CSF but also seem to leave the CSF compartment to undergo further maturation in the periphery. Although a direct proof for B cell trafficking patterns would require in vivo B cell tracking, several studies show that B cells in various tissues, such as cervical lymph nodes [52], connect with CNS B cells, with maturation steps occurring on both sides of the BBB. A schematic summary of B cell trafficking is displayed in Figure 2.Combined analyses of CSF transcriptome and proteome B cell repertoires further reveal that intrathecal Ig is indeed produced by CSF B cells/B cell clones.

However, the main obstacle, in order to fully dissect MS B cell pathophysiology, is the lack of confirmed B cell target antigens. As a possible explanation, it has to be considered that CSF/CNS B cells are not specific for a single antigen but may target numerous antigens exposed during tissue injury. Thus, persistent CSF B cell maturation possibly reflects ongoing tissue injury in MS and a chronic immune response to such immunologic stimulus. NGS data collectively suggest that B cells encounter antigen(s) within the CSF and traffic across the BBB, with further maturation taking place in the periphery, possibly feeding autoimmune mechanisms through the interaction with other immune cells. Regarding autoantibody associated neuro-inflammation/encephalitis, only a limited number of studies with low patient numbers are available yet. Nevertheless, these studies, in combination with control patients in MS studies, show that intrathecal B cell maturation and B cell trafficking across the BBB also occurs in diseases outside of MS.

Altogether, NGS could be established as a powerful tool to use when studying B cell trafficking and maturation between different compartments, which helps to understand B cell-related pathologies within the CNS. One major open question regarding B cell-mediated autoimmunity comprises the characterization of initial events that possibly start an autoimmune cascade, as well as the mechanisms which sustain chronic inflammatory processes. Finally, the assessment of treatment-specific effects on B cell repertoires provide further insights in B cell-mediated (patho-)physiology and in return, improve our understanding of drug-specific (side) effects and modes of action, finally leading towards more personalized medicine.

## 5. Outlook

Several improvements have recently been achieved for NGS B cell repertoire generation and analysis. Unique molecular identifiers (UMIs) have been increasingly used in order to avoid over-amplification and sequencing errors during the generation of immunoglobulin transcript libraries. Ideally, the UMI is implemented during the reverse transcriptase step to directly mark Ig transcripts, before further amplification steps are carried out. In addition, common bioinformatics pipelines evolved during the last few years, allowing better data quality and comparability between different studies. In addition, sub-sampling approaches [88] have recently been applied to take into account that a different sequencing depth might occur between samples. With the mentioned improvements, targeted NGS sequencing has been well established for the characterization of B cell responses under different conditions.

As another highly interesting sequencing approach, single-cell sequencing with platforms similar to the 10× Genomics Chromium, is rapidly emerging [89]. This method allows to not only sequence Ig transcripts from single B cells but also to assess the “whole transcriptome” data from one particular cell. However, specific B cell analysis pipelines might not be as elaborate, as in NGS platforms. Currently, only very limited data is available on CSF whole transcriptome analyses. An overall analysis of CSF lymphocytes was recently performed in two multiple sclerosis studies [90,91] with one study also analyzing B cell repertoires [91]; however, these studies mostly concentrated on changes in transcriptional pathways. Additionally, single-cell next-generation sequencing (10X Genomics) was also applied to study CSF signatures of COVID-19-infected patients [92]. There was a striking increase in the proportion of B cells in the CSF and monoclonal antibodies generated from these expanded clusters that were reactive against COVID-19 antigens. As already shown in peripheral blood and cell cultures [93], computational pipelines will also enable deep CSF B cell clonotyping on a single-cell level in the near future.

## Figures and Tables

**Figure 1 diagnostics-11-01871-f001:**
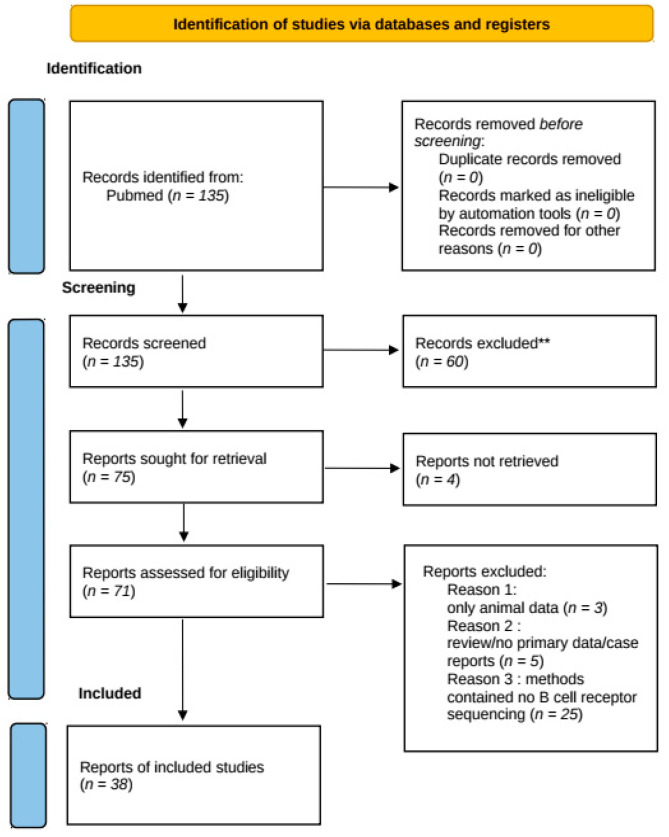
Workflow for the identification of studies. ** Records excluded after screening of abstract due to non-neurological purposes such as lymphoma, glaucoma or leukemia.

**Figure 2 diagnostics-11-01871-f002:**
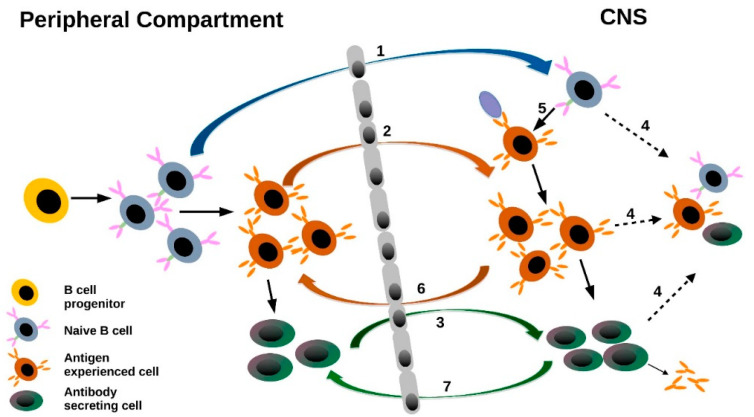
Schematic overview of B cell trafficking via the blood–brain barrier. Possible scenarios: (1) influx of naïve B cells, (2) influx of antigen-experienced B cells, (3) influx of antibody-producing B cells, (4) longitudinal persistence, (5) intrathecal activation, (6) efflux of antibody-experienced B cells and (7) efflux of antibody-producing B cells.

**Table 1 diagnostics-11-01871-t001:** B cell repertoires analysis in MS patients.

Study	Diagnosis/Number of Patients	Compartment of B Cell Analysis	Methods	VH Family Bias (within CSF) Towards:	CSF Clonal Expansion (in MS Patients)	Clones between CNS/CSF and Periphery
(Qin et al., 1998)	MS/*n* = 12 OND/*n* = 15	CSF	tRT-PCR IgVH-PCR	VH4	• Dominant clone(s) (10/12 MS patients) • Numerous SHM	N/A
(Colombo et al., 2000)	MS/*n* = 10 OND/*n* = 10	CSF/blood	tRT-PCR IgVH-PCR	VH3 VH4	• Oligoclonal B cell accumulations (10/10 MS patients)	CSF B cells hardly represented in PB, compartmentalized clonal expansion within the CSF
(Owens et al., 2003)	MS/*n* = 4 Viral meningitis/*n* = 2	CSF	ScRT-PCR, IgH/L-PCR	N/A	• B cell clonal IgG expansion (3/4 MS patients)	N/A
(Owens et al., 2007)	MS/*n* = 15 OND/n = 2	CSF/blood	ScRT-PCR IgH/L-PCR	VH4-39 VH4-31 VH4-59	• 64% of CSF CD138 Cells in clonal populations	N/A
(Bennett et al., 2008)	CIS/*n* = 10	CSF/blood	ScRT-PCR IgH/L-PCR	VH4 VH2 (70%)	• Expanded B and plasma cell clonal populations	N/A
(H.-Christian von Büdingen et al., 2012)	MS/*n* = 6 OND/*n* = 7	CSF/blood	NGS	IGHV4-39 IGHV4-59 IGHV4-61	• CSF-restricted B cell activation in MS patients	• Bidirectional exchange across the BBB in a restricted pool of clonally related B cells • Clusters undergo active diversification primarily in the CNS, in the periphery or in both compartments in parallel
(Palanichamy et al., 2014)	MS/*n* = 8	CSF/blood	NGS	IGHV4-39 IGHV4-59 VH4-61	• B cells belonging to bicompartmental clusters may have been exposed to antigen-stimulation in the CSF or PB	• SM: most frequent immune axis between PB and CSF • Class-switched DN B cells: also clonally related to CSF Ig repertoires • PB plasma cells: few bicompartmental clusters
(Beltrán et al., 2014)	MS/*n* = 12 OND/*n* = 7	CSF/blood	High-throughput pyrosequencing	VH4-34, 4-39; VH4-59; VH4-4, 4-61; VH4-31	• Extensive SHM in CSF IgM antibodies • AICDA expressed in CSF IgM-producing B cells	Clonal tracking/lineage: • Ancestors of CSF B cell clones reside in PB • Maturation continues in CSF • No isotype switching from IgM to IgG in CSF
(Stern et al., 2014)	MS/*n* = 5	CNS tissue, CLN	SSC/NGS	IGHV4 (CNS vs. CLN) IGHJ usage biased toward IGHJ4 in CNS and CLNs	• Class switched, acquired SHM and expanded clones in CNS B cells	• Both less mature and more experienced offspring observed in the CNS and CLN • Maturation steps not restricted to a single compartment • Clonal expansion of B cells possibly occurs in multiple compartments
(Johansen et al., 2015)	MS/*n* = 10 OIND/*n* = 6	CSF/blood	NGS	IGH-V4	• Higher frequency of IGHV4 genes and more replacement mutations in CSF of MS patients	Dominant B-cell clones • produce CSF IgG • are present at both sides of BBB • Go through several rounds of mutations in the CSF
(Eggers et al., 2017)	MS = 11 (bulk NGS)	CSF/blood	NGS	N/A	• ASC are clonally expanded in the CSF • ASC participate in production of clonal CSF IgG	• Clonal relationships between CSF and PB B cells • Migration of B cells and activation in the CNS in active MS • Clonal relationships between CSF and PB suggest influx of functionally diverse B cells
(Greenfield et al., 2019)	RRMS = 8 PPMS = 2	CSF/blood	NGS	N/A	• Exclusive CSF IgG-VH clusters (10/10 patients) • Exclusive CSF IgM-VH (9/10) • mixed IgM and IgG clusters (5/10)	• Clonal connections between PB and CSF in 10/10 patients
(Kowarik et al., 2020)	MS = 8	CSF/blood	NGS	VH4 (at baseline)	• In CSF 80% of VH sequences within clonal populations at baseline	Lineage analyses of clonal groups: • Bidirectional exchange across the BBB • M, DN and plasmablasts from PB contribute 30% to clonal groups emanating from PB, naïve B cells < 10%

Abbreviations: tRT-PCR, RT-PCR from total RNA from CSF cells; scRT-PCR, single-cell RT-PCR from CSF cells; IgVH-PCR, PCR amplification of Ig heavy chains variable region; IgH/L-PCR, PCR amplification of Ig heavy/light chains; Ig-VH region, Ig heavy chains variable regions; SM, class-switched memory B cells; M, memory B cells; DN, double negative (germinal center-like) B cells; ASC, antibody secreting cells; PB, peripheral blood; CSF, cerebrospinal fluid; CLN, cervical lymph node; NGS, next-generation sequencing; SSC, single-cell next-generation sequencing; BBB, blood–brain barrier.

**Table 2 diagnostics-11-01871-t002:** Clonal persistence over time.

Source	Patient	Treatment between T1 + T2	Time between T1 + T2 (Months)	Overlapping CSF-Clones T1–T2
Tomescu-Baciu et al., 2019	MS1	Natalizumab	18	37
	MS2	Interferon-beta 1a	18	24
Greenfield et al., 2019	1	Fingolimod	14	8
	2	Interferon	15	9
	3	Dimethyl fumarate	22	1
	4	Natlizumab	13	2
	5	-	12	1
	6	Interferon	13	-
	7	Fingolimod	15	-
	8	Natalizumab	9	-
	9	-	15	-
	10	-	13	-
Kowarik et al., 2020	S1	Natalizumab	6	1
	S5	Natalizumab	6	5
	S6	Natalizumab	6	5
	S7	Natalizumab	6	0
	S2	Fingolimod	6	0
	S3	Fingolimod	6	-
	S4	Fingolimod	6	0
	S8	Fingolimod	6	0

## Data Availability

Not applicable.

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
