# Peer review of "Next Generation Sequencing of Cerebrospinal Fluid B Cell Repertoires in Multiple Sclerosis and Other Neuro-Inflammatory Diseases—A Comprehensive Review"

_diagnostics, 2021, doi:10.3390/diagnostics11101871_

Round 1

Reviewer 1 Report

A nice review on actual topic.

The manuscript includes recent data of the role of intratecal B-cells in pathogenesis of autoimmune inflammatory diseases of the brain.
It includes discussion of possible mechanisms of B-cell entrance into CSF and the data on their activity inside the BBB, molecular mechanisms of their participation in mechanisms of different diseases.
the minor recommendation is to add data of the role of B-cells in forming the intra meningeal follicles, typical for progressing MS. These follicles mainly consist of B-cells and theit role in immunoregulation must be discussed.
After adding this the manuscript could be accepted for publication in special issue on MS and other demyelinating diseases, which is under preparation nom.

Author Response

We thank the reviewer for the thoughtful comments and further include the topic of B cell associated follicles in the meninges which will add to the general topic of B cells and CNS immunity.

Page 8, results:

As mentioned above, clonally expanded B cells could not only be detected in the CSF and parenchymal infiltrates but also in meningeal lymph-like follicles/aggregates of mainly progressive MS patients [73]. Interestingly, the majority of expanded anti-gen-experienced B cells derived from the meninges were also present in the parenchyma. Altogether, clonal relationships could be established between different compartments including the meninges, inflammatory MS plaques and the CSF compartment.

Reviewer 2 Report

Dr. Ruschil and his colleagues have put together a comprehensive review describing the involvement of sequencing technologies on B cell repertoires in inflammatory diseases.

This is an important topic and clinically relevant. However, the review could be significantly improved.

  1. The objective and conclusion of the review is not clearly indicated in the abstract
  2. The Introduction should be reorganized. This section is too long and does not reflect the aim of the review. The key argument is not precisely stated. It is also important to explain the organization of the review in this section.
  3. The body of the review should be also reorganized. It is crucial to create links between the discussed research findings and the research question. It will establish a more coherent review article.
  4. The discussion should be more concise and clearly summarize the important aspects of the existing literature.
  5. The quality of figure 2 is low and too schematic. Furthermore, the Figure is not mentioned in the manuscript and its importance is not clear.
  6. To avoid grammar and syntax errors, English should be professionally edited.

Author Response

Following the suggestions of reviewer 2, we substantially revised our manuscript, reorder the manuscript in several parts and also tried to provide a broader frame for the subject-specific findings of our review.

1. The objective and conclusion of the review is not clearly indicated in the abstract.

We thank the reviewer for this suggestion and added the following sentences to make the abstract make the abstract more concise.

Page 1, Abstract:

In particular B cell maturation and trafficking across the blood-brain-barrier (BBB) has recently been deciphered with the development of next generation sequencing (NGS) approaches that allow the assessment of representative cerebrospinal fluid (CSF) and peripheral blood B cell repertoires.

In conclusion, these data suggest an intense exchange of B cells across the BBB possibly feeding auto-immune circuits.

2. The Introduction should be reorganized. This section is too long and does not reflect the aim of the review. The key argument is not precisely stated. It is also important to explain the organization of the review in this section.

According to this and the following comments we understand, that the structure of the review has to be further explained and specified at certain points. Following the suggestion of reviewer 2, we shortened the introduction at several passages (especially in regard to next generation sequencing methods 1.3.).

In addition to the abstract, we also clearly state the major aim of the review which includes the characterization of B cell migration and development. We also explain the organization of this review (page 2, introduction 1.1):

With the development of new high through-put sequencing techniques, there have been huge advances in tracking B cell development and trafficking by the assessment of B cell repertoires in the CSF and peripheral blood compartment on a transcriptome level. The aim of the review is to summarize these developments and extract the major findings on B cell maturation and migration in neuro-inflammatory diseases. We first provide a short background on the biological context of B cell sequencing and the approach of repertoire sequencing and then explain the methods of this review. In the following, we will summarize major studies on B cell receptor sequencing in a chronological order and discuss the major findings of the mentioned studies.

3. The body of the review should be also reorganized. It is crucial to create links between the discussed research findings and the research question. It will establish a more coherent review article.

We thank the reviewer for this comment and add a short passage that again explains the structure of the results (page 5, results):

In order to learn more about B cell maturation and trafficking in neuro-inflammatory diseases, we performed the above mentioned search strategy and found several studies on B cell repertoire mass sequencing in multiple sclerosis and single studies on NMOSD as well as LGI1 and NMDAR autoantibody mediated encephalitis. Early studies in MS with conventional B cell repertoire sequencing in the CSF compartment were also included in the analysis to compare between the different methods and complete the picture of B cell pathophysiology in MS. B cell repertoire mass sequencing results in complex data sets for which reasons we ordered the results according to findings from basic repertoire analysis, clonal expansion within the CSF, B cell trafficking across the BBB and –if available- longitudinal CSF studies on single MS patients.

We also added the following sentences in the results sections to further structure the results and create links for the discussion.

Page 6, results:

The basic analysis of B cell repertoire properties in the CSF of MS patients indicates,….

Page 8, results:

By applying NGS sequencing, it has become possible to representatively sequence both – CSF and peripheral blood B cell repertoires and thus examine B cell trafficking across the BBB. Overall, a substantial clonal overlap between peripheral blood and CSF B cells could be established in different studies (Table 1).

4. The discussion should be more concise and clearly summarize the important aspects of the existing literature.

According to the suggestion of reviewer2 we reorganized the discussion and changed the following statements to make the discussion more concise:

I. Regarding the analysis of basic B cell repertoire characteristics, a preferential usage of the VH4 germline family within CSF B cell repertoires has been consistently shown in several studies.

III. The most striking results derived from NGS studies on CSF and PB B cell repertoires relate to B cell trafficking across the BBB. An intense exchange of B cells across the BBB is observed in MS and other autoimmune disorders pointing towards an involvement of additional compartments outside of the CNS. Studies consistently show, that B cells not only migrate into the CSF but also seem to leave the CSF compartment to undergo further maturation in the periphery. Although a direct proof for B cell trafficking patterns would require in-vivo B cell tracking, several studies show that B cells in various tissues such as cervical lymph nodes [52] connect with CSF B cells with maturation steps occurring on both sides of the BBB. A schematic summary of B cell trafficking is displayed in Figure 2.

One major open question regarding B cell mediated auto-immunity comprises the characterization of initial events that possibly start an auto-immune cascade as well as the mechanisms and compartments which sustain chronic inflammatory processes. Finally, the assessment of treatment specific effects on B cell repertoires provides further insights in B cell mediated (patho-)physiology and in return improves our understanding of drug specific (side) effects and modes of action, finally leading towards a more personalized medicine.

5.The quality of figure 2 is low and too schematic. Furthermore, the Figure is not mentioned in the manuscript and its importance is not clear.

We thank the reviewer for this suggestions and heavily modified Figure 2 to make the statement of a bidirectional exchange of B cells across the BBB clear. We also give a short legend on different B cell trafficking patterns and relate to the figure in the discussion section.

We improved the quality of the embedded figure. If the resolution is still too low, we could provide the figure separately (e. g. as .pdf)

6. To avoid grammar and syntax errors, English should be professionally edited

The manuscript has been additionally proof-read by a native speaker.

Round 2

Reviewer 2 Report

The revised manuscript has been greatly strengthened by the changes to the text. No further comments for the authors.